# GSAP: A Global Structure Attention Pooling Method for Graph-Based Visual Place Recognition

**Yukun Yang [1], Bo Ma [1,\*], Xiangdong Liu [1], Liang Zhao [2] and Shoudong Huang [2]**

[1]    Beijing Institute of Technology, Beijing 100081, China; 3120150384@bit.edu.cn (Y.Y.); xdliu@bit.edu.cn (X.L.)
[2]    Centre for Autonomous Systems (CAS), Faculty of Engineering and Information Technology, University of Technology Sydney, Ultimo, NSW 2007, Australia; Liang.Zhao@uts.edu.au (L.Z.); Shoudong.Huang@uts.edu.au (S.H.)
\*    Correspondence: bma000@bit.edu.cn

**Abstract:** The Visual Place Recognition problem aims to use an image to recognize the location that has been visited before. In most of the scenes revisited, the appearance and view are drastically different. Most previous works focus on the 2-D image-based deep learning method. However, the convolutional features are not robust enough to the challenging scenes mentioned above. In this paper, in order to take advantage of the information that helps the Visual Place Recognition task in these challenging scenes, we propose a new graph construction approach to extract the useful information from an RGB image and a depth image and fuse them in graph data. Then, we deal with the Visual Place Recognition problem as a graph classification problem. We propose a new Global Pooling method—Global Structure Attention Pooling (GSAP), which improves the classification accuracy by improving the expression ability of the Global Pooling component. The experiments show that our GSAP method improves the accuracy of graph classification by approximately 2–5%, the graph construction method improves the accuracy of graph classification by approximately 4–6%, and that the whole Visual Place Recognition model is robust to appearance change and view change.

**Keywords:** graph construction; graph neural networks; graph convolution; graph global pooling; visual place recognition

## 1. Introduction

With the development of robotics and computer vision in recent years, improvement in the accuracy of localization and mapping is urgently needed. Given a sequence of images captured from different places, the images of the same place should be found, which is the Visual Place Recognition (VPR) problem [1]. VPR is a key component of image-based Localization, Mapping, and Simultaneous Localization and Mapping (SLAM). Because VPR can help reduce the accumulative error in the applications mentioned above, it has attracted more attention in recent years. It is a challenging task for the following three reasons (the first two are shown in Figure 1):

- The viewpoint of the same place can change drastically when the place is revisited.
- The appearance can change due to the illumination and seasonal change.
- Having a large number of images in the database causes high computational cost.

Most of the previous research focuses on the image processing of the VPR system. Early studies (e.g., Bag of Words (BoW)-based [2]) use traditional descriptors such as SIFT or SURF, while it is not necessarily effective when appearance changes. Artificial intelligence has made great progress in recent years, with most researches focusing on deep learning methods [3–5]. Most of the VPR studies use convolutional features [6–13]. Chen et al. design a framework using convolutional features and untrainable pooling layers [8]. Arandjelovic et al. design a trainable pooling layer to improve the performance. However, convolutional features are not robust enough on appearance and viewpoint

change [6]. In addition, semantic segmentation is introduced to construct a graph [14,15]. As semantic graph expression is invariant in the first two challenging scenarios, Gawel et al. use a random walk graph kernel method to extract descriptors in the semantic graph [14]. However, random walk is a kind of graph kernel method, and good performance of graph kernel methods usually comes at the cost of heavy computational consumption. Furthermore, the semantic segmentation results are not fully utilized when constructing graph data in Abel's study [14].

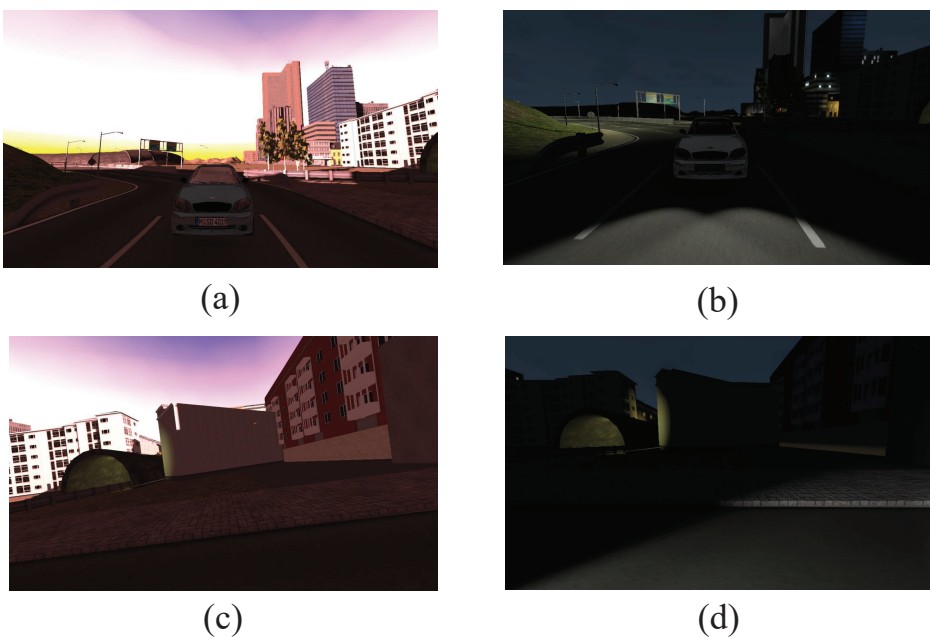

(a)  (b)

(c)  (d)

**Figure 1.** A difficult scene in the SYNTHIA dataset for the Visual Place Recognition (VPR) problem. Subfigures (**a–d**) are taken from one place in different views, when they have different appearance (e.g., illumination, weather or seasonal change, or all), or both: Viewpoint changes between subfigures (a–d). Appearance changes between subfigures (a–d). Viewpoint and appearance both change between subfigures (a,d), which is the most challenging scene.

To solve the appearance change and viewpoint change problem, we try to extract more robust features. We first transform the image sequence to graph sequence by using results of semantic segmentation and depth image, i.e., we transform the place recognition to graph classification problem and we fuse the semantic and geometric features that are robust to appearance and view changes. As the Graph Neural Network (GNN) has shown advantages in dealing with graph data [16], we use the GNN model to complete the classification task, and we propose a novel graph Global Pooling method to improve the classification accuracy. The training of the constructed graph data is more time-consuming than training on the original image data directly. Because the most commonly used Graph Global Pooling is not injective, it cannot map distinct multisets of node features into unique embeddings [17]. This can lead to false positive results. We design a trainable global pooling layer to improve the expression ability, though the injective property is still not guaranteed.

The contributions of this paper are as follows.

- Based on the graph construction approach in X-View [14], we propose a graph data construction approach that transforms the RGB image and the depth image into the graph for place recognition by extracting both semantic and geometry information.
- To improve the expression ability of the GNN architecture, we propose a Global Pool method—Global Structure Attention Pooling. Compared with the most commonly used global pooling methods, e.g., global mean pooling, global max pooling, and

global sum pooling, our pooling method is a trainable pooling method improving the expression ability. Compared with other trainable pooling methods, our work not only catches the first-order neighbor information by learning attention scores, but also models the spatial relation by the Gate Recurrent Unit (GRU) for higher-order neighbor information.

The rest of the paper is organized as follows. Some related work and their similarities and differences with the VPR problem are presented in Section 2. The methodology of the proposed graph construction method and GSAP is presented in Section 3. Experiment results are shown in Section 4 and followed by discussions in Section 5. The conclusion is drawn in Section 6.

## 2. Related Work

In this section, we review some literatures related to the components of our proposed VPR architecture, including VPR research works based on different kinds of feature or information, image retrieval, and graph classification.

### 2.1. Visual Place Recognition

As cameras are the primary sensors of autonomous systems, visual place recognition has been attracting increasingly more attention, aiming to choose loop closure candidates for the SLAM algorithm. There are two main challenges to perform visual place recognition based on the differences of scenes: one is the appearance change caused by illumination condition and seasonal changes, and the other is the viewpoint change caused by revisiting one place from different viewpoints [1]. In the VPR literature, various feature extraction methods have been developed for visual place recognition, including deep convolutional feature-based methods [6–13], handicraft feature-based methods [2,18], semantic information-based methods [19–25], sequence-based methods [26,27], and graph-based methods [19,20,28–32]. Overall, most of these studies focus on the image processing module of the visual place recognition system, which aims to extract and describe features that are robust in the different challenge conditions as mentioned above. Compared with these graph-based methods, our work also concentrates on the extraction and representation of global features. However, the novelty lies in that we are the first to promote the VPR problem by improving the expression ability of Global Graph Pooling. Our model can recognize more complex patterns by learning structural information and property information alternately. Compared with these convolutional-based methods, our work is more robust in drastically view changing and appearance changing scenes.

### 2.2. Image Retrieval

With the wide spread of the Internet and search engines, efficient and accurate image retrieval has greatly progressed in recent years, including class-level and instance-level image retrieval. Given an image of an instance, other images of this instance in the database should be found, which is the aim of instance-level image retrieval. If the given instance is an image of a certain place and the database contains a sequence of images *w.r.t.* the place, it becomes a VPR problem. Most VPR problems are considered retrieval problems as well. In the literature, various methods have been developed for image retrieval, including text-based methods, content-based methods [33–38], sketch-based methods [39–41], and semantic-based methods [36,41,42]. Overall, most of the studies focus on feature extraction and representation. Place recognition can be treated as a classification problem as well [8].

### 2.3. Graph Classification

In recent years, we have witnessed the success of Graph Neural Networks (GNNs) in modeling complex and irregular data. Specifically, the Graph Convolutional Network [43] (GCN) together with other GNN variants, e.g., Graph Attention Networks [44] (GAT), GraphSAGE [45], Relational Graph Convolutional Networks [46] (R-GCNs), and Graph Isomorphism Network [17] (GIN), have been proposed for neighborhood aggregation.

Furthermore, some general representations of GNN have also been summarized, e.g., Message Passing Neural Network [47] (MPNN), Non-local Neural Network [48] (NLNN), and Graph Networks [49] (GN).

Graph classification is a graph-level learning task, which concentrates on global information. A common architecture is used to fuse the local information and then fuse the global information. To fuse local information, Hierarchical Pooling is generally utilized, e.g., TopKPooling [50,51], SAGPooling [52], EdgePooling [53], ASAPooling [54], etc.

After the Hierarchical Pooling layer, a Global Pooling layer is designed to get the global embedding. Such mechanisms can aggregate all the nodes at one time and get a fixed length global representation. A sum, mean, and max function is often utilized as Global Pooling layer. However, the expression ability of such layers is not enough, which means different graph features may have the same representation and result in false classification. Structural information is completely lost using these kinds of Global Pooling layers. To improve the variousness of global representation and classification accuracy, some Global Pooling methods are studied [55–57]. The Recurrent Neural Network (RNN) achieves increasing attention in modeling sequence data [58,59]. In this paper, we use edges to compute the score for node features, get a feature sequence, and then extract global representation by GRU [60], which can fuse the structural information and node information together in global representation.

## 3. Methodology

In this section, we present the proposed graph-based VPR model. First, an overview of the VPR pipeline is given. Second, we present the proposed graph construction approach in detail. Finally, the details of the GSAP method are described, the process after Global Pooling is described briefly, and the loss function used in our approach is presented.

### 3.1. Overview of the VPR Pipeline

An overview of the proposed VPR pipeline is shown in Figure 2. We use semantic segmentation and depth image pairs to construct graph data, which are done off-line. The detailed process of graph construction is presented in Section 3.2. The graph data are fed into a GNN model to get the graph embedding. Then, the graph embedding is used in a Multilayer Perception (MLP) to get the classification results. In our GNN model, we use GIN [17] as Graph Convolution layer to aggregate the node features. After this, we apply Batch Normalization (BN) over a batch of node features [61]. Then, we use a Global Pooling (GP) layer to learn the global representation. The reason why we do not adopt Hierarchical Pooling, e.g., SAGPooling [52] or EdgePooling [53], is that introducing Hierarchical Pooling cannot improve the performance of graph classification. Actually, in most GNN architectures, the convolutional layer can quickly lead to smooth node representations [62].

The Graph Convolution layer (GIN) updates node representations as follows:

$$\mathbf{x}'_i = h_\Theta \left( (1 + \epsilon) \cdot \mathbf{x}_i + \sum_{j \in \mathcal{N}(i)} \mathbf{x}_j \right) \tag{1}$$

Here, $h_\Theta$ denotes a neural network. We use MLP in this paper. $\epsilon$ can be a parameter to be learned or a fixed scalar. $\mathbf{x}_i$ is the node representation of the $i-$th node. $\mathcal{N}(i)$ is the set of nodes adjacent to $i$. $\mathbf{x}_j$ represents the neighbor node of $\mathbf{x}_i$. $\mathbf{x}'_i$ is the node representation of $\mathbf{x}_i$ in the next layer.

We propose a novel Global Pooling method—Global Structure Attention Pooling. The details are described in Section 3.3.

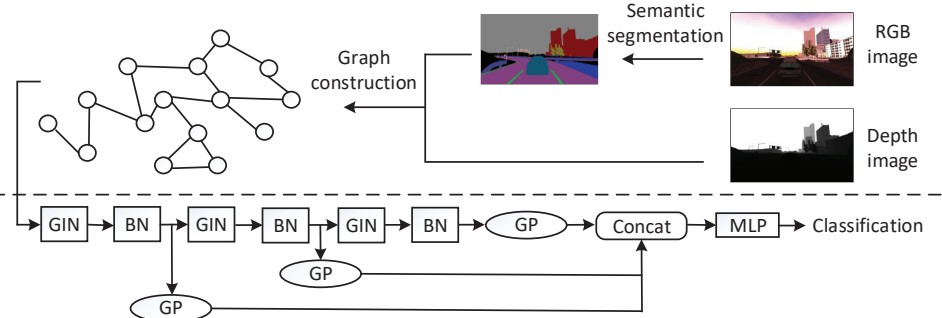

**Figure 2.** The architecture of our Visual Place Recognition (VPR) pipeline. The top half is the process of graph construction. The bottom half is our network architecture. We consider the Graph Isomorphism Network (GIN), Batch Normalization (BN), and Global Pooling (GP) layer as a unit which is piled up three times. The three outputs are concatenated as the graph global embedding.

*3.2. Graph Construction*

Based on Gawel et al.'s graph construction approach [14], we proposed a graph construction approach for the 3-D case. In our approach, the geometric and semantic information is utilized to reserve more information that contributes to the following classification. We define a graph $\mathcal{G} = (\mathcal{V}, \mathcal{E}, \mathcal{X})$, where $\mathcal{V}, \mathcal{E}, \mathcal{X}$ are the set of nodes, edges, and node features, respectively. The workflow is listed in Algorithm 1. First, we get the semantic segmentation and depth image corresponding to the RGB image, which is the same as Gawel et al. [14] do in their work. Second, we extract semantic labels and blob attributes from the semantic segmentation result and depth image. Finally, the undirected graph is assembled as follows:

---

**Algorithm 1** Graph Construction

---

**Input:** RGB image $I$, depth image $D$
**Output:** constructed graph $\mathcal{G} = (\mathcal{V}, \mathcal{E}, \mathcal{X})$
 1: compute semantic segmentation results $S$
 2: extract blobs in $S$
 3: blobs $\rightarrow \mathcal{V}$
 4: **for** each blob **do**
 5:     compute $u, v, x, y, w, h, a$
 6:     find node label in $S$
 7:     find the depth corresponding to $(u, v)$ in $D$
 8:     compute $(X, Y, Z)$
 9:     $(onehot(label), X, Y, Z, x, y, w, h, a) \rightarrow \mathbf{x}, \mathbf{x} \in \mathcal{X}$
10: **end for**
11: **for** every two blobs **do**
12:     $b_1$ bitwise or $b_2 \rightarrow b_{or}$
13:     compute $N$ in $b_{or}$
14:     compute $d_e$
15:     **if** $d_e < d_t$ and $N = 2$ **then**
16:         edge connected, edge $\in \mathcal{E}$
17:     **else**
18:         do nothing
19:     **end if**
20: **end for**

---

3.2.1. Nodes Determination and Node Labels

Every blob is regarded as a node. Every blob has a corresponding semantic label. We regard the semantic label as the graph label.

### 3.2.2. Node Attributes

The node attributes include 8 elements: $X, Y, Z, x, y, w, h, a$.
Where $(X, Y, Z)$ is the 3-D location expressed in camera frame. The last 5 elements are blob attributes: $(x, y)$ is the top left corner coordinate of the external polygon of a blob, and $w, h, a$ are the width, height, and area of the external polygon of a blob, respectively.

Given the semantic segmentation result, the last 5 elements can be computed by using openCV. The first three elements are derived as follows.

By using the depth image, every node has its corresponding depth and pixel location. We transform this information to the 3-D location in the camera frame with a camera projection model [63]:

$$\begin{pmatrix} X \\ Y \\ Z \end{pmatrix} = Z \begin{pmatrix} f_x & 0 & c_x \\ 0 & f_y & c_y \\ 0 & 0 & 1 \end{pmatrix}^{-1} \begin{pmatrix} u \\ v \\ 1 \end{pmatrix} \tag{2}$$

where $(u, v)$ is the pixel coordinates, and $f_x, f_y, c_x, c_y$ are camera intrinsics.

### 3.2.3. Edges Determination

The edges are connected by the blobs' proximity and their 3-D distances. The edges are without labels and attributes. We find the proximate blobs by bitwise or operation of every two blobs $b_1$, $b_2$ and we can get $b_{or}$. After that, we compute the number $N$ of connected components of $b_{or}$. If $N = 2$, $b_1$ and $b_2$ are proximate blobs. To exclude the false neighbors caused by the shelter, we also consider the Euclidean distance $d_e$ between $(X_{b1}, Y_{b1}, Z_{b1})$ and $(X_{b2}, Y_{b2}, Z_{b2})$ which are the locations of $b_1$ and $b_2$. If $d_e$ is smaller than the threshold $d_t$, $b_1$ and $b_2$ are not false neighbors. Overall, if $N = 2$ and $d_e < d_t$, the edge between these two nodes is connected. Conversely, there is no edge between these two nodes.

### 3.2.4. Node Features

We combine the node label and node attributes together as node feature. Thus, the input node feature $\mathbf{x}_i$ of our GNN architecture is

$$\mathbf{x}_i = \begin{pmatrix} onehot(label) & X_i & Y_i & Z_i & x_i & y_i & w_i & h_i & a_i \end{pmatrix} \tag{3}$$

where $onehot(label)$ is the node label one hot encoding of node $i$.

In this way, every image is transformed into a graph.

### 3.3. Global Structure Attention Pooling

Expression ability has been widely researched in recent GNN-related studies [17]. We design a Global Structure Attention Pooling (GSAP) method to improve the expression ability of the Global Pooling layer and the performance of graph classification. The basic process of GSAP is shown in Figure 3 and the detailed process is as follows.

In general, edges and nodes in one graph have relation with each other. Considering that edges in graphs contain structural information, we compute a score for each edge by its corresponding two node features. We concatenate the two features and obtain the score of every edge with a single full connection layer with LeakyReLU activation function. The score of the edge between node $i$ and node $j$ is computed by the following equation:

$$\alpha_{ij} = Leaky\,ReLU(\mathbf{w}^T(\mathbf{x}_i \parallel \mathbf{x}_j) + b), \ \mathbf{x}_i, \ \mathbf{x}_j \in \mathbb{R}^n \tag{4}$$

where $\parallel$ represents concatenation, $\mathbf{w}$ and $b$ are the parameters that need to be learned, $n$ is the dimension of $\mathbf{x}_i$, and $\mathbf{x}_j$.

Then, the node score is computed according to its connected edges:

$$s_i = \sum_{j \in \mathcal{N}(i)} e^{\alpha_{ij}} \tag{5}$$

where $\mathcal{N}(i)$ is the set of nodes adjacent to $i$.

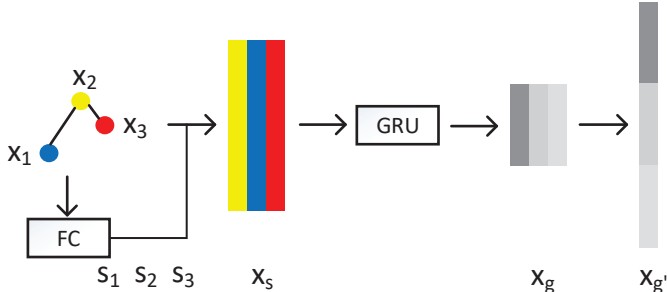

**Figure 3.** The Global Structure Attention Pooling (GSAP) process. Qualitatively, we assume that the graph has three nodes.

Given node scores, the feature sequence fuses node information and structural information together. The node feature $\mathbf{x}_i$ is weighted by its node score $s_i$. As graphs may have a different number of nodes, the length of the feature sequences may be different. We add $\mathbf{0}$ vector in the end to keep the sequence to a certain length. Then, we get the feature sequence $\mathbf{x}_s$ as follows:

$$\mathbf{x}_s = \begin{pmatrix} s_1\mathbf{x}_1 & s_2\mathbf{x}_2 & ... & s_i\mathbf{x}_i & ... & s_k\mathbf{x}_k & \mathbf{0} & ... \end{pmatrix} \tag{6}$$

where $k$ is the number of nodes in the graph. $\mathbf{0}$ is zero vector whose dimension is $n$ as well. The number of $\mathbf{0}$ is an adjustable parameter $m$. In order to reserve all the information of a feature graph, $k$ and $m$ should satisfy the following condition:

$$k + m \geq k_{max} \tag{7}$$

where $k_{max}$ is the maximal number of nodes in all graphs.

Even if we get the sequence of $\mathbf{x}_s$ by considering the first-order neighbor of each node, the nodes without connection to each other can also have a certain relation. GRU can model the information of several consecutive nodes with its reset gate and update gate. We utilize GRU to extract the feature graph global descriptor $\mathbf{x}_g$.

$$\mathbf{x}_g = GRU(\mathbf{x}_s), \ \mathbf{x}_s \in \mathbb{R}^{n \times (k+m)} \tag{8}$$

where $\mathbf{x}_g$, $\mathbf{x}_g \in \mathbb{R}^{p \times (k+m)}$ is a padded concatenation of every GRU output.

Finally, we reshape matrix $\mathbf{x}_g$ into a vector $\mathbf{x}'_g$:

$$\mathbf{x}'_g = reshape(\mathbf{x}_g) \tag{9}$$

Let $q = p \times (k + m)$, then $\mathbf{x}'_g \in \mathbb{R}^q$. In our GNNs, we concatenate the three feature graph global descriptors $\mathbf{x}'_{g1}$, $\mathbf{x}'_{g2}$, and $\mathbf{x}'_{g3}$. Thus, the graph global descriptor for classification is

$$\mathbf{x}_{gd} = \big\|_{i=1}^{3} \ \mathbf{x}'_{gi} \tag{10}$$

$\mathbf{x}_{gd}$, $\mathbf{x}_{gd} \in \mathbb{R}^{3q}$ is passed into MLP layer, followed by a Log Softmax function to generate a probability distribution over all classes and compute its logarithm for numerical stability, and then we compute NLL loss to optimize the parameters of the network.

## 4. Results

### *4.1. Datasets*

We use two original datasets to prepare their corresponding graph datasets: Airsim dataset [14] and SYNTHIA dataset [64], in which RGB images, depth images, semantic segmentation labels, and odometry information are provided.

### 4.1.1. Airsim

This is a simulated dataset made by a photo-realistic Airsim framework [14]. Images of top-down view and forward-facing view (as shown in Figure 4) are collected by an Unmanned Aerial Vehicle and a car. Each of these view sequences contains over 5000 images with associated ground truth. Here, the semantic classes are misc, street, building, car, sign, fence, hedge, tree, wall, bench, power line, rock, and pool. For each view sequence, useful information is provided, such as ground truth for semantic segmentation, instance segmentation, global camera poses, depth images, and calibration parameters. The viewpoint change between each top-down and forward-facing view pair is 90°. The class is balanced as the camera motion is uniform.

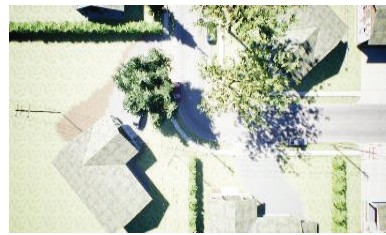 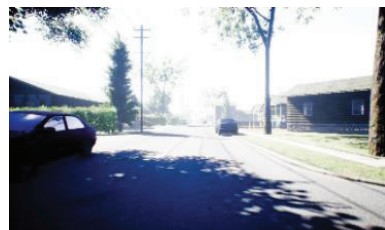

Forward view            Downward view

**Figure 4.** The samples of forward view and downward view images in the Airsim dataset. They are sampled from the same location.

We use the waypoint files to sample images at a constant distance of around 10 m that contains 50 image frames. We deal with the forward-facing trajectory in the same way. Ignoring the offset in *z* direction, the forward view and its corresponding downward view are given the same label when constructing graph label file. Every class has 100 graphs in total. We construct the node and edge level information by the approach described in Section 3.2. Some statistical data of our Airsim graph dataset are shown in Table 1. The number of nodes reflects the richness of the semantic information. The number of edges reflects the complexity of the graph structure. Whether the class is balanced has an effect on the choice of evaluation metrics.

**Table 1.** Statistics of our graph datasets.

| Statistics | Airsim | SYNTHIA |
|---|---|---|
| Number of Graphs | 10,000 | 10,479 |
| Number of Classes | 100 | 89 |
| Class Imbalance | No | Yes |
| Average of Nodes per Graph | 27.19 | 68.60 |
| Average of Edges per Graph | 64.11 | 88.58 |

### 4.1.2. SYNTHIA

Different from the Airsim dataset, each subsequence in the SYNTHIA dataset consists of the same traffic situation but under different weather, illumination, and season conditions, and the class is imbalanced as the camera motions are not uniform. The current subsequences are Spring, Summer, Fall, Winter, Rain, Soft-rain, Sunset, Fog, Night, and Dawn. Each of these subsequences contains approximately 8000 images with associated ground truth. For each subsequence, useful information is provided, such as 8 views,

ground truth for semantic segmentation, instance segmentation, global camera poses, depth images, and calibration parameters. Here, the semantic classes are misc, sky, building, road, side walk, fence, vegetation, pole, car, sign, pedestrian, cyclist, and lane-marking.

We use global camera locations to label image frames every 20 m. As shown in Figure 5, we choose the forward and leftward views to make sure the viewpoint change is 90°, and we choose four kinds of appearances, i.e., Dawn, Night, Summer, and Fog. The key steps of node and edge construction are the same as Airsim. The statistical data of the SYNTHIA graph datasets are shown in Table 1 as well. The same places in different subsequences have different appearances and views.

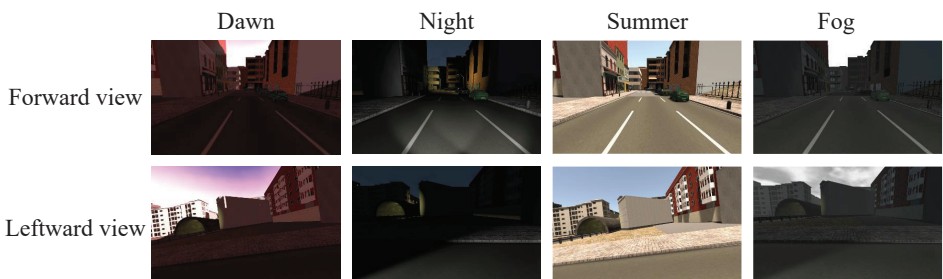

**Figure 5.** The samples of the forward view and leftward view images with four different appearances: Dawn, Night, Summer, and Fog in the SYNTHIA dataset. They are sampled from the same location.

### 4.2. Evaluation Metrics

For the graph classification task, we utilize various evaluation metrics from the previous classification work. For the experiments using the Airsim dataset, we aim to measure the contribution of the key components in our GNN model. We use Accuracy ($Accuracy = \frac{Correcct\ Prediction\ Number}{Total\ Sample\ Number}$) as the evaluation metric, as the Airsim dataset has a balanced class. We use **Training Accuracy** to measure the expression ability of the GP methods and use **Test Accuracy** to measure the generalization ability.

For the experiments using the SYNTHIA dataset, we aim to compare the performance of our VPR model with other VPR methods. We remove the MLP after training, compute the embeddings of all the graphs in test set, and then compute the **Euclidean distances** of every two embeddings. We show the performances via the **Precision–Recall Curve** as the class in SYNTHIA dataset is imbalanced.

### 4.3. Task Setting

For the experiments in Section 5, the Airsim dataset is divided into three parts with a ratio of 6:2:2, the data of two sequences are mixed together, namely, the training set, validation set, and test set.

For the experiments in Section 4.5, the forward sequence and downward sequence are divided into three parts with a ratio of 6:2:2, respectively. Here, we use the same random seed of the random split function for different sequences to make sure the train, validation, and test sets have no geographical overlap. The first and second parts of the downward sequence are considered as the training set and validation set, respectively. The third part of the forward sequence is considered as the test set.

As for the SYNTHIA dataset, the four subsequences are divided in the same way as mentioned above for the Airsim dataset. The first and second parts of the "Dawn" subsequence are considered as the training set and validation set, respectively. The third part of the "Night", "Summer", and "Fog" subsequences is considered as the test set, respectively.

We use AdapNet [65] as a semantic segmentation net. First, we train AdapNet [65] on the training set. Second, we use AdapNet [65] and Algorithm 1 to obtain the graph data of all the datasets. Third, we train the GNN model by using the graph data. Finally, the classification test results can be obtained.

### 4.4. Methods to Compare

The hyperparameters for our model and all the models to be compared are searched in a certain range as shown in Table 2. We consider the following VPR methods as comparison to our VPR model in the SYNTHIA dataset.

**Table 2.** The hyperparameters of our model and compared models. The hidden size is only for Graph Neural Network (GNN) models. The 0 value of Weight decay is used for the experiments on training set to show the expression ability of the GNN models. The number of nodes reserved is only for the model with GSAP.

| Hyperparameter | Range |
| --- | --- |
| Learning rate | $1 \times 10^{-4}, 5 \times 10^{-3}, 1 \times 10^{-3}, 1 \times 10^{-2}$ |
| Hidden size | 128, 256, 512 |
| Weight decay | $0, 1 \times 10^{-2}, 1 \times 10^{-3}, 1 \times 10^{-4}$ |
| Number of nodes reserved | 50–100 |

- NetVLAD [7]: This is a deep learning method that combines Locally Aggregated Descriptors (VLAD) with Convolutional Neural Networks. For a fair comparison, we use MLP to classify after getting the global descriptor, and the training and test processes are also the same as our VPR method.
- AMOSNet [8]: This is also a deep leaning method using a 2-D image. It uses a convolution kernel, and the feature map is fed into two fully connected layers.
- DBoW2 [2]: It extracts the handicraft features, generates the dictionary by clustering these features, and looks for the corresponding words of a query image in the dictionary.

We consider the following Global Pooling methods to be compared with our GSAP methods in the Airsim dataset.

- Global Add Pooling (GDP): It returns batch-wise graph-level outputs by adding node features across the node dimension.
- Global Mean Pooling (GMP): It returns batch-wise graph-level outputs by averaging node features across the node dimension.
- Global Max Pooling (GAP): It returns batch-wise graph-level outputs by taking the channel-wise maximum across the node dimension.
- Set2Set [55]: Based on iterative content-based attention, it has the property that the vector retrieved from our memory would not change if we randomly shuffled the memory to output sequences.

The abstracted information in the graph construction step has an effect on the following task. We conduct the experiments in the Airsim dataset to compare the different kinds of graph construction approaches:

- Semantic Information based [14]: It extracts semantic labels, and instances center 3-D locations as graph features.
- Geometric Information based: We evaluate the effectiveness of the geometric information separately. It extracts the instance center 3-D location; the top left corner coordinate of the external polygon of a blob; and the width, height, and area of the external polygon of a blob as graph feature.

### 4.5. Results and Analysis

We present the results of all the VPR comparison methods in the SYNTHIA dataset in Figure 6. First, our VPR model performs the best among all the compared VPR models; it ensured a relatively high Precision Rate when Recall Rate becomes larger. The Precision Rate becomes very low when the Recall Rate is close to 1. On one hand, this could be because the overlap between two adjacent frames is relatively high, i.e., they are on the class boundary. As the data we use are image sequences, the gap between two proximate

frames with different classes is too small to distinguish. On the other hand, some images of the same class with different views do not have overlap with each other, which is another possible reason. Second, the 2-D image deep learning-based method NetVLAD does not perform better than our VPR model. This could be because it cannot utilize the 3-D information. Besides, it relies on the scene appearance and viewpoint. In our experiments, the data in the test set have very different appearances and views compared with the training set. NetVLAD is not robust to these kinds of changes, which leads to a bad performance generalizing to different scenes. Third, AMOSNet performs worse than NetVLAD. It uses similar convolutional layers for feature map generation. Different from AMOSNet, NetVLAD has a trainable pooling layer to learn the crusting centers of local features (as shown in Table 3), which leads to the generation of more effective image representations. Finally, DBoW2 gives a worse performance as the handicraft feature is not robust enough compared with the graph feature and deep convolutional feature.

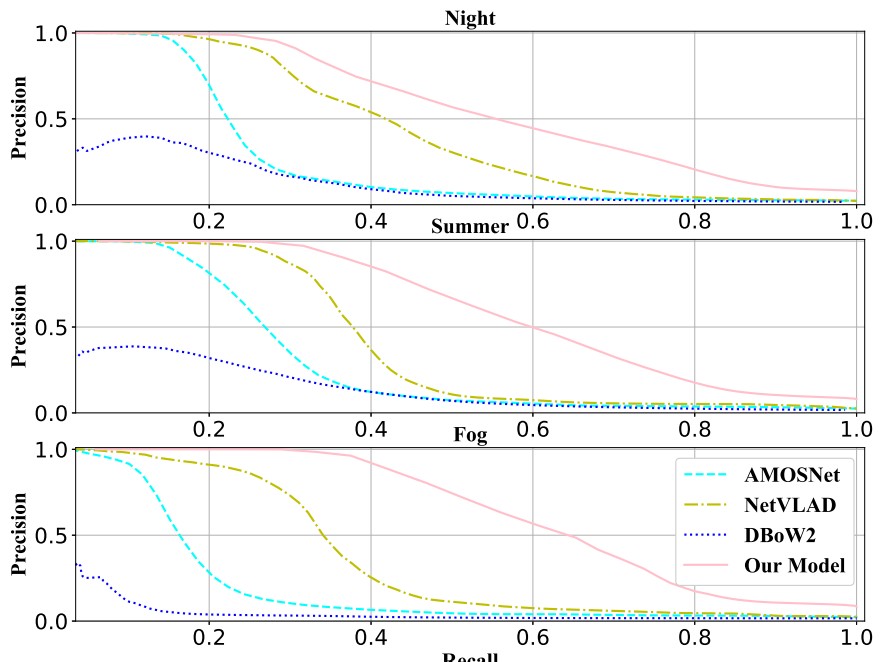

**Figure 6.** The Precision–Recall Curve comparison of the VPR models in subsequences "Night", "Summer", and "Fog" of SYNTHIA dataset.

**Table 3.** The comparison and component analysis of the VPR models.

| Method | Component |
| :---: | :---: |
| Our model | Graph construction + GSAPs |
| NetVLAD | 2D image + deep feature + trainable pooling |
| AMOSNet | 2D image + deep feature |
| DBoW2 | 2D image + handicraft feature |

We present the results of all the VPR comparison methods in the Airsim dataset as well. As shown in Figure 7, compared with the results in the SYNTHIA dataset, our VPR model also performs the best among all the compared VPR models. However, AMOSNet performs better than NetVLAD in the Airsim dataset, which means that AMOSNet is better at recognizing the viewpoint change scene. The trainable pooling layer of NetVLAD can improve the expression ability, but strong expression ability may lead to weak generation ability sometimes. Thus, expression ability and generation ability need to be balanced.

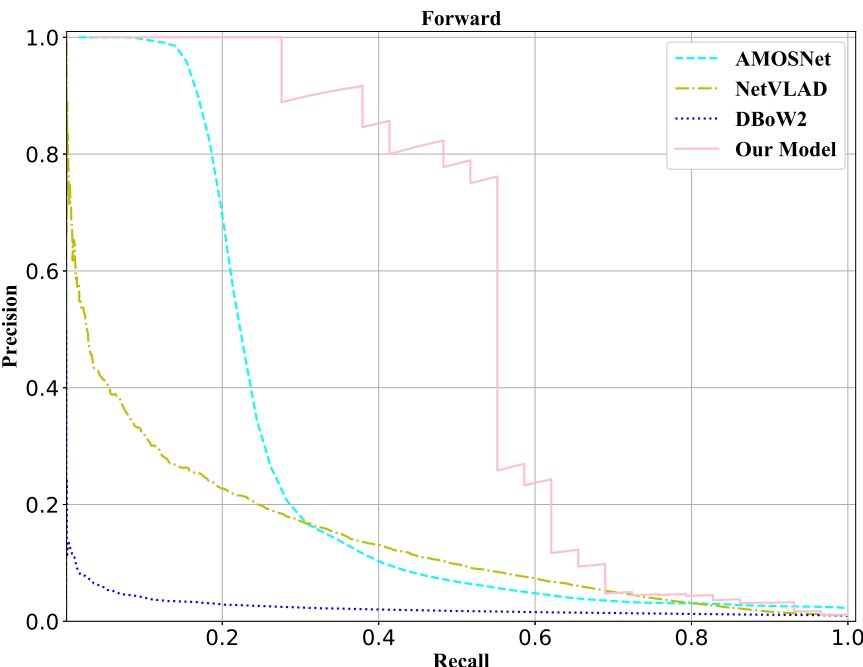

**Figure 7.** The Precision–Recall Curve comparison of the VPR models in "Forward view" sequence of the Airsim dataset.

By summarizing all the results, we can see that our GNN-based model is competitive for solving the VPR task on both the SYNTHIA and Airsim datasets, especially when the view and appearance change drastically.

## 5. Discussions

### 5.1. Effect of Structure Attention Pooling Method

We conduct the experiments in the Airsim dataset in order to compare the performances of Global Pooling methods. Global Pooling is a necessary component in graph classification. In our GNN model, we propose GSAP to get the global descriptor. We first examine the effect of GSAP by comparing with other Global Pooling methods. We compare them by using classification accuracy. The expression ability can be measured in the training dataset. Figure 8 shows that GSAP reaches the highest accuracy using the least epochs and its curve is the most stable without sudden change. GDP, GMP, and Set2set have similar performance. GAP performance the worst. As for generalization ability, it shows that our GSAP method achieves the best accuracy among these methods in the upper half of Table 4. Structural information can help improve the classification accuracy.

**Table 4.** The results of different global pooling methods and graph construction methods on Airsim test set. Here, the performances are presented by average accuracy and standard deviation of 10 random seeds of model initialization parameters.

| Global Pooling Methods | Accuracy |
|---|---|
| GDP | $91.485 \pm 0.716$ |
| GMP | $91.315 \pm 1.255$ |
| GAP | $90.270 \pm 0.567$ |
| Set2Set | $88.570 \pm 1.314$ |
| Our GSAP | $\mathbf{93.575 \pm 0.699}$ |

| Graph Construction Methods | Accuracy |
|---|---|
| Semantic based | $87.630 \pm 0.361$ |
| Geometric based | $90.063 \pm 1.892$ |
| Semantic and geometric based | $\mathbf{93.575 \pm 0.699}$ |

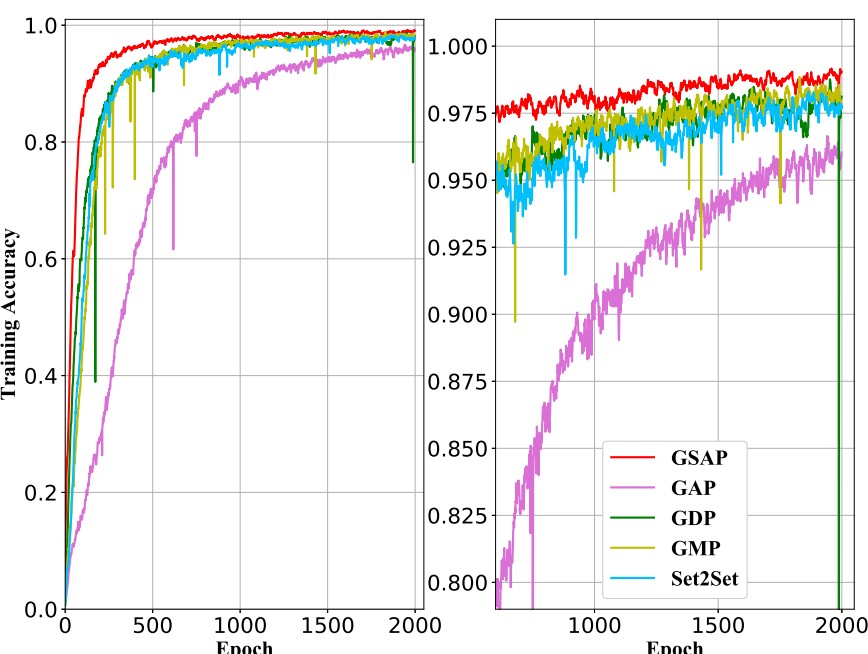

**Figure 8.** The Training Accuracy results. The right half is a partial enlargement of the left half. This figure reflects the expression ability of the compared global pooling methods. The higher the training accuracy they achieve, the stronger expression ability they have.

### 5.2. Effect of Graph Construction

We conduct the experiments in Airsim to compare and evaluate the effectiveness of the graph construction approaches. By using different construction approaches shown in Section 4.4, we get different graph data. Results in the lower half of Table 4 show that a relatively high accuracy is achieved when just relying on geometric information, but the distribution of values is relatively dispersed. Semantic information contributes to a more stable performance but the accuracy values are relatively low. Our graph construction approach integrates and reserves more effective information for the graph classification task, which leads to the best performance. The main difficulty when applying it is that the graph construction would be time-consuming if computing resources are limited.

### 6. Conclusions

In this paper, we transform the VPR problem into a graph classification task, and then we use the GNN model to solve the VPR problem. In our data preparation task, we propose a graph construction approach that extracts core information for the classification task. In our GNN model, we design a Global Pooling method by transforming graph features to a sequence and predicting the global representation by GRU. We conduct extensive experiments in different appearances and view scenes to verify the effectiveness and robustness of our VPR model. We can conclude that the expression ability improvement of GNNs can contribute to the graph classification performance. The proposed method outperforms the state-of-art VPR algorithms in terms of the Precision rate and Recall rate. The limitation is that the graph construction would be time-consuming if the computing resource is limited.

In our current work, the graph construction is done off-line. The whole architecture is not end to end. In future work, we will consider improving the edges construction approach by link prediction to make the constructed data more suitable for the following classification task and so that the whole network can be trained end to end.

**Author Contributions:** Methodology, experiments and manuscript writing Y.Y.; experiments improvement guidance and review, B.M.; supervision and funding acquisition, X.L.; language modifi-

cation and review, L.Z. and S.H. All authors have read and agreed to the published version of the manuscript.

**Funding:** This research received no external funding.

**Conflicts of Interest:** The authors declare no conflict of interest.

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
