# Peer review of "GSAP: A Global Structure Attention Pooling Method for Graph-Based Visual Place Recognition"

_remotesensing, doi:10.3390/rs13081467_

Round 1

Reviewer 1 Report

The work proposes a novel method for Visual Place Recognition (VPR) by transforming the VPR problem on a graph classification problem using Graph Neural Networks (GNN), which outperformed convolutional-based methods. The method was evaluated on two datasets and compared with three other literature methods, achieving higher results than the comparisons.

The work took inspiration from Abel Gawel et al. (X-View) approach, proposing enhancements, but X-View itself was not directly compared. Another noteworthy point is that every accuracy value reported for the Airsim dataset on Table 2 is higher than the accuracy reported in Abel Gawel et al. (85%). What can be the reason behind that? Was the dataset evaluated differently? I understand that adding a comparison of X-View with your method would significantly improve the quality of the experiments and show the impact of GSAP method.

Equation 9 presents a reshaping function, but what is the specific operation being made? Is it a padded concatenation of every GRU output or other method? Because the cited work for learning sequence representations from RNNs [51] uses the last output to compute the representation. This can be made more explicit.

The overall study was well conducted, the methodological soundness is high and the results are clear. However, besides the points raised above, several typos in the text should be addressed in more careful reading. I pointed some of those typos and plural concordance issues below:

  1. improve -> improves

15., 28. year -> years

  1. Further more -> Furthermore
  2. has showed ->  has shown
  3. preseanted -> presented
  4. these graph based method -> methods
  5. wide spread -> widespread
  6. imasge -> image
  7. the graph global descriptor are -> is
  8. divided into threes -> three
  9. when generalize -> generalizing
  10. to improve -> improving

Reviewer 2 Report

The paper presents a Graph Attention Pooling based Visual Place Recognition (VPR) method using RGB and depth information. Two synthetic datasets are used where one is used for classification (its relation to VPR not clear though) and the other for VPR. For the latter, comparisons are done against three representative state-of-the-art (sota) approaches. Several ablations are provided to justify and understand design choices. Source code is not promised.

*Novelty and Claims*

The authors’ claimed contributions with respect to a modified usage of RGB (semantics) + depth as compared to [11] when constructing graphs and use of Global Structure Attention Pooling (GSAP) to create a novel descriptor type for VPR. The authors also claim, “we are the first to introduce GNN based architecture to solve VPR problem”, which is not completely true, please see [a,b,c] and revise/narrow this particular claim accordingly.

*Method*

The authors have explained their method well and the design choices of pooling type and inclusion of input data modality are justified through ablation studies.

*Datasets*

Only two datasets have been used and both of them are synthetic. Furthermore, only one of them is used to present results on VPR including sota comparisons. This is one of the main weaknesses of the paper. The authors should include real datasets, e.g. Oxford Robotcar (as in [19] which also uses semantics-based method).

*Implementation Details and Data Splits*

The authors have not provided implementation details of their training procedure. The statement, “We use two of the sub-sequences to construct training set and the other two to construct test set.” seems to mean that there is a geographical overlap between training and testing set. This is an incorrect setting for the task of VPR. Train, val and test splits should not overlap geographically. The authors are recommended to include clear split details perhaps through a plot. Furthermore, in Section 4.3, the authors should clarify how a reference set and query set is defined for each of the train/val/test splits.

*Experiments*

Table 1 is not well explained. What is the concept of “class” here? What do you mean by “number of graphs”? Is it a geographical division of the whole traverse? Are you casting VPR as a classification problem? What do you mean by “class imbalance”? This makes it hard to interpret results on the AirSim dataset.

*Results and Ablations*

Ablations are good but the main results are sparse as each dataset is only used for a specific task.

Overall, the presented method seems to be promising and relevant to the VPR community. Experiments and results are however weak. Please address the concerns raised above to improve the quality of the manuscript.

Minor flaws:

Fig 1 - dataset name is missing

L31 - Citation style doesn’t seem to be correct (e.g. Abel Gawel should ideally be just Gawel)

L44 - Graph pooling not being injective is not properly explained/referenced

L118 - Since the authors use depth information, it might be useful to include a subsection in Related work on use of depth for VPR

L138 - “learn-able”

[a] Zhang, Xiwu, et al. "Graph-based place recognition in image sequences with CNN features." Journal of Intelligent & Robotic Systems 95.2 (2019): 389-403.

[b] Sun, Qi, et al. "Dagc: Employing dual attention and graph convolution for point cloud based place recognition." Proceedings of the 2020 International Conference on Multimedia Retrieval. 2020.

[c] Kong, Xin, et al. "Semantic Graph Based Place Recognition for 3D Point Clouds." IROS 2020

Reviewer 3 Report

Decision:  Major Revision

Comments: The idea is good and interested in graph-based visual place recognition but the authors did not present it well. The authors are suggested to keenly focus on the cohesion and coherence of the manuscript writing. I recommend some major and minor edits, which should be addressed in the revised version carefully with details.

  1. The abstract should be consistent with the main text of the paper’s content. The abstract must be concise. Your abstract is incomplete and not well concise as well as some sentences are incomplete, for the sentence see sentences 1, 2, and 3, etc.  I will strongly recommend to re-write the abstract as we as avoid repetition and the authors should add numerical improvements in the abstract against the state-of-the-art. For assistance see “DOI: 10.3390/s20010183” and cite behind the CNN to enrich the literature.
  2. More suitable keywords should be selected and I strongly recommend all keywords should be sorted in alphabetical order.
  3. In the first paragraph of the introduction, section authors need to explain the background of the proposed work. The current version of the paper lack explanatory details about this domain. I strongly recommended explaining some background of the proposed work in the first paragraph.
  4. Extension to introduction will be appreciated: clstm: deep feature-based speech emotion recognition using the hierarchical convlstm network, clustering based speech emotion recognition by incorporating learned features and deep bilstm.
  5. The major defect of this study is the debate or Argument is not clearly stated in the introduction session. Hence, the contribution is weak in this manuscript. I would suggest the author to enhance your theoretical discussion and arrives at your debate or argument.
  6. Current challenges are not clearly mentioned in the introduction section of this paper. I suggest adding a paragraph about the current challenges of this area followed by the authors’ contribution to overcoming those challenges.
  1. The manuscript, however, does not link well with recent literature on recognition that appeared in relevant top-tier journals, e.g., the IEEE Intelligent Systems department on " att-net: Enhanced emotion recognition system using lightweight self-attention module". Also, new trends of AI for recognition “mlt-dnet: recognition using 1D dilated CNN based on multi-learning trick approach” are missing it should be comprised.
  2. A complete framework with each step explained visually is missing from the paper. Figure 2 is just a working flow without any visual contents to show the proposed system in action for an input image advanced to the final output. I strongly recommended adding a complete self-explanatory framework, which shows each step visually.
  3. What is the main difficulty when applying the proposed method? The authors should clearly state the limitations of the proposed method in practical applications.
  4. The datasets are explained well, but I recommend adding some visual representation of a few samples from each dataset.
  1. The validation parameters are general, need to workout innovative parameters on validations. The performance of the proposed model must present with appropriate parameters.
  2. The comparative analysis with state-of-the-art is missing the system needs to campare with recent SOTA methods and show a dedicated table in the revised version for system comparisons and I strongly recommend more experiments with a different aspect to show the model's significance.
  3. Please make sure and revise your conclusions section underscore the scientific value-added of your paper, and/or the applicability of your findings/results, as indicated previously. Please revise your conclusion part into more details. Basically, you should enhance your contributions, limitations, underscore the scientific value-added of your paper, and/or the applicability of your findings/results and future study in this session.
  4. Tables and picture formatting in the experimental results section are very poor. Especially the text inside figures need consistency. Captions given to them are not self-explanatory.
  5. Figure 5 should be explained in more detail.
  6. The paper needs extensive proofreading, there are some grammatical mistakes and typos in the current version, I recommend proofread and revise the paper carefully.

Looking forward to seeing you with the revised version.

Round 2

Reviewer 2 Report

The authors have addressed the concerns raised earlier, including clearly stating the problem formulation (VPR as a classification problem), clarifying the experimental setup in terms of train/val/test splits and evaluation, and providing additional results. 

Reviewer 3 Report

The authors successfully addressed my Comments and suggestions.